# Synthetic Analogues of Gibbilimbol B Induce Bioenergetic Damage and Calcium Imbalance in *Trypanosoma cruzi*

**DOI:** 10.3390/life13030663

**Published:** 2023-02-28

**Authors:** Maiara Amaral, Marina T. Varela, Ravi Kant, Myron Christodoulides, João Paulo S. Fernandes, Andre G. Tempone

**Affiliations:** 1Instituto de Medicina Tropical, Faculdade de Medicina, Universidade de São Paulo, São Paulo 05403-000, Brazil; 2Centre for Parasitology and Mycology, Instituto Adolfo Lutz, São Paulo 01246-000, Brazil; 3Departamento de Ciências Farmacêuticas, Universidade Federal de São Paulo, Diadema 09913-030, Brazil; 4Molecular Microbiology, School of Clinical and Experimental Sciences, Faculty of Medicine, University of Southampton, Southampton SO16 6YD, UK; 5Medical Biotechnology Laboratory, Dr. B.R. Ambedkar Center for Biomedical Research, University of Delhi, Delhi 110021, India

**Keywords:** *Trypanosoma cruzi*, natural products, gibbilimbol, mechanism of action, drugs, treatment

## Abstract

Chagas disease is an endemic tropical disease caused by the protozoan *Trypanosoma cruzi*, which affects around 7 million people worldwide, mostly in development countries. The treatment relies on only two available drugs, with severe adverse effects and a limited efficacy. Therefore, the search for new therapies is a legitimate need. Within this context, our group reported the anti-*Trypanosoma cruzi* activity of gibbilimbol B, a natural alkylphenol isolated from the plant *Piper malacophyllum*. Two synthetic derivatives, LINS03018 (1) and LINS03024 (2), demonstrated a higher antiparasitic potency and were selected for mechanism of action investigations. Our studies revealed no alterations in the plasma membrane potential, but a rapid alkalinization of the acidocalcisomes. Nevertheless, compound 1 exhibit a pronounced effect in the bioenergetics metabolism, with a mitochondrial impairment and consequent decrease in ATP and reactive oxygen species (ROS) levels. Compound 2 only depolarized the mitochondrial membrane potential, with no interferences in the respiratory chain. Additionally, no macrophages response of nitric oxide (NO) was observed in both compounds. Noteworthy, simple structure modifications in these derivatives induced significant differences in their lethal effects. Thus, this work reinforces the importance of the mechanism of action investigations at the early phases of drug discovery and support further developments of the series.

## 1. Introduction

Chagas disease (CD) is a parasitic infection caused by *Trypanosoma cruzi* that triggers ~14,000 deaths annually and prevails in Latin America. Recently, CD has also spread worldwide due to the immigration of infected people, which is concerning for non-endemic countries [1]. The lifecycle of *T. cruzi* comprises a mammalian host and an insect vector, a triatomine bug, besides distinct morphological forms adapted to different microenvironments. Epimastigotes, a non-infectious parasite form, proliferate in the vector midgut and, when reach the hindgut, differentiate into infective trypomastigotes. Triatomines excrete this trypomastigotes in their feces, infecting the mammalian host through skin lesions or mucous membranes. Once inside the host, the parasites invade macrophages and other nucleated cells, transforming into intracellular amastigotes, which replicate by binary division and differentiate back into trypomastigotes, which rupture the cell membrane and invade new cells. The cycle is completed when a new insect vector ingests the parasites during a blood meal [2].

This disease is characterized by two different clinical phases: the acute phase, mostly asymptomatic and marked by the presence of high blood trypomastigotes parasitemia, and the chronic phase, typified by cardiomyopathy and cardiac impairment, with 30–40% of patients developing lethal conditions. However, the clinically relevant form of the parasite is the intracellular amastigotes, which is found in the chronic phase of the disease. Trypomastigotes are also of importance for drug discovery. Once they are found in humans at the acute phase and if not treated, it can lead to the reactivation of the parasitemia and disease transmission [3]. Although many groups are in the search for more effective and well-tolerated drugs against CD, no new therapies have reached the market in the past 50 years. Benznidazole and nifurtimox are the only available drugs, but present several limitations due to poor efficacy in the chronic phase of the disease and severe adverse effects, resulting in a low compliance of patients. These issues highlight the need for new efficacious and safer drugs [3,4].

With an extensive chemical diversity, natural products formed the basis of the first drugs discovered and have served as an inspiration for approximately 50% of all FDA-approved drugs [5]. These substances derived from plants, animals, and microorganisms are present in nature in great variety and abundance, resulting in a substantial diversity of secondary metabolites that can be produced. Therefore, natural products are a useful source of novel and unique molecular entities, enabling the identification of new therapeutic targets and unraveling new lethal antiparasitic mechanisms. They also play a key role as prototypes for new chemical molecules with improved pharmacological potential and properties. However, unlike traditional drugs, the isolation and characterization of products from natural source can be time-consuming and some of them may have a complex chemical structure, making it difficult to synthesize and obtain them on a large scale [6,7]. Gibbilimbol B, an alkylphenol isolated from *Piper malacophyllum*, has been reported to present an antiparasitic activity against trypomastigotes, the infective form of *T. cruzi*, with a target in the parasite cell membrane [8]. Because of their structural simplicity, the synthesis of this compound is easily accessible, allowing for the functionalization and addition of new groups. In this context, in a hit expansion campaign from our working group, a new series (named LINS03) was designed, using the natural compound gibbilimbol B as a scaffold. Among the synthesized compounds, compound LINS03018 (**1**) and LINS03024 (**2**) (Figure 1) showed a higher potency and selectivity against trypomastigotes and intracellular amastigotes (Table 1). Furthermore, mechanistic assays of **1** and **2** in trypomastigotes demonstrated no plasma membrane damage, a typical effect induced by the natural prototype [9,10].

Assessing the function of some essential organelles and pathways is a feasible approach to unveil the mechanisms involved in the effects of these new molecular entities as well as the strength of their potential as lead compounds. Therefore, in this work, the effects of **1** and **2** were investigated on plasmatic and mitochondrial membrane potentials, adenosine triphosphate (ATP), reactive oxygen species (ROS), and calcium intracellular levels, as well as acidocalcisomes. In addition, their hemolytic and immunomodulatory effects were also assessed. A protein modelling, binding site identification, and molecular docking study complete with an analysis of crucial interactions, was also done for these compounds with one potential target enzyme. These valuable results helped to understand the mechanisms elicited by these compounds that alter the homeostasis of the parasite.

## 2. Materials and Methods

### 2.1. Chemicals and General Procedures

Compounds 4-methoxi-N-octylbenzamide (**1**) and 1-phenyldec-1-en-3-ol (**2**) were prepared as previously reported, with an adequate purity level (>95%) to the study [9,10]. DiSBAC2(3) (Bisoxonol), JC-1, H2DCFDA, ATP Determination Kit, acridine orange and fluo-4 AM were obtained from Molecular Probes^®^ (Invitrogen™,Grand Island, NY, USA). Fetal bovine serum (FBS) was purchased from Gibco and all other reagents/culture media were acquired from Sigma-Aldrich (Saint Louis, MO, USA).

The absorbance, fluorescence- and luminescence-based assays were evaluated using the spectrofluorimeter/luminometer FilterMax F5 Multi-Mode Microplate Reader (Molecular Devices, San Jose, CA, USA) or the cytometer Atunne NxT flow (Thermo Fisher Scientific, Waltham, MA USA).

### 2.2. Hemolytic Activity

Erythrocytes (3%) obtained from BALB/c mice were seeded in 96-U-shaped microplate and incubated with compounds **1** and **2** serially diluted in PBS from 6.5 to 200 µM, at 24 °C for 2 h. The hemolytic activity was determined in the cell supernatants using a spectrophotometer at 570 nm [12]. Milli-Q ultrapure water was employed as a positive control and untreated erythrocytes as a negative control.

### 2.3. Parasites and Mammalian Cells

*T. cruzi* (Y strain) trypomastigotes were maintained in LLC-MK2 cells (ATCC CCL 7) and cultivated with RPMI-1640 medium and 2% FBS, at 37°C in a 5% CO_2_ humidified incubator. Bone-marrow-derived macrophages were acquired from BALB/c mice femurs washed with RPMI-1640 medium supplemented with 10% NCTC cell (ATCC clone L929) supernatant and 20% FBS. The cells were maintained at 37°C in a 5% CO_2_ humidified incubator for approximately seven days [13]. The procedures were performed according to the Animal Use Ethics Committee (AGT-CEUA 05/2018).

### 2.4. Mechanism of Action (MoA) Studies

To establish the standard conditions, trypomastigotes (2 × 10^6^/well) were seeded in 96-well plates and incubated with compounds **1** and **2** serially diluted in HBSS medium supplemented with D-glucose (10 mM) from 1.6 to 300 µM, at 37 °C for 3 h in a 5% CO_2_ humidified incubator. Parasite viability was determined by the resazurin colorimetric method [14]. Untreated parasites were employed as a negative control.

### 2.5. Plasma Membrane Electric Potential (Δψ_p_)

Trypomastigotes (2 × 10^6^/well) were incubated with compounds **1** (230.2 µM) and **2** (32.06 µM) in HBSS medium supplemented with D-glucose (10 mM) for 3 h. Then, bisoxonol (0.2 µM) was added for 5 min and the fluorescence was monitored in a flow cytometer with excitation 488 nm and emission 574 nm (BL-2) [15]. Gramicidin (0.5 µg/mL) was employed as a positive control and untreated parasites as a negative control [16].

### 2.6. Mitochondrial Membrane Electric Potential (ΔΨ_m_)

Trypomastigotes (2 × 10^6^/well) were incubated with compounds **1** (230.2 µM) and **2** (32.06 µM) in HBSS medium supplemented with D-glucose (10 mM) for 3 h. Subsequently, JC-1 (10 µM) was added for 20 min and the fluorescence was monitored in a flow cytometer with excitation of 488 nm and emission of 530 (BL-1)/574 nm (BL-2). The mitochondrial membrane potential was analyzed by the ratio between channels BL-2 and BL-1 [17]. CCCP (100 µM) was employed as a positive control and untreated parasites as the negative control.

### 2.7. Adenosine Triphosphate (ATP)

Trypomastigotes (2 × 10^6^/well) were incubated with compounds **1** (230.2 µM) and **2** (32.06 µM) in HBSS medium supplemented with D-glucose (10 mM) for 3 h. In sequence, 0.5% Triton X-100 was added for lysis and parasites were mixed with ATP Determination Kit reaction buffer (1 mM DTT, 1.25 µg/mL luciferase and 0.5 mM luciferin), following the manufacturer instructions. Luminescence was monitored in a spectroluminometer and an ATP curve (1 to 6000 nM) was used as the standard [18]. CCCP (25 µM) was employed as the positive control and untreated parasites as the negative control.

### 2.8. Reactive Oxygen Species (ROS)

Trypomastigotes (2 × 10^6^/well) were incubated with compounds **1** (230.2 µM) and **2** (32.06 µM) in HBSS medium supplemented with D-glucose (10 mM) for 3 h. Later, H_2_DCFDA (5 µM) was added for 15 min and the fluorescence was monitored in spectrofluorimeter with excitation 485 nm and emission 535 nm [19]. Sodium azide (10 mM) was employed as positive control and untreated parasites as negative control.

### 2.9. Intracellular Calcium (Ca^2+^)

Trypomastigotes (2 × 10^6^/well) were stained with Fluo-4 AM (5 µM) in PBS for 1 h. The parasites were then incubated with compounds **1** (230.2 µM) and **2** (32.06 µM) and the fluorescence was monitored every 20 min up to 3 h in a spectrofluorimeter with excitation of 485 nm and emission of 535 nm [20]. 0.5% (*v*/*v*) Triton X 100 was employed as the positive control and untreated parasites as the negative control.

### 2.10. Acidocalcisomes

Trypomastigotes (2 × 10^6^ /well) were stained with acridine orange (4 µM) in PBS for 5 min. After this period, the parasites were incubated with compounds **1** (230.2 µM) and **2** (32.06 µM) and the fluorescence was monitored every 20 min up to 3 h in a spectrofluorimeter with excitation of 485 nm and emission of 535 nm [21]. Nigericin (4 µM) was employed as the positive control and untreated parasites as the negative control.

### 2.11. Immunomodulatory Response

Bone marrow-derived macrophages (5 × 10^5^ /well) were seeded in 24-well plates and incubated overnight at 37 °C with 5% CO_2_. The cells were infected with trypomastigotes (10:1 amastigotes/macrophage) for 2 h and then treated with compounds **1** and **2** serially diluted from 30 to 3.75 µM in RPMI-1640 medium and 2% FBS. After 48 h, the supernatants were collected and analyzed with a Griess reagent in a spectrophotometer with 570 nm [22,23]. A NaNO_2_ curve (0 to 400 µM) was used as the standard. LPS (25 µg/mL) was employed as the positive control and untreated cells as the negative control [2].

### 2.12. Protein Modelling, Binding Site Identification, Molecular Docking Studies, and Analyses of Crucial Interactions

The 492 long amino acid sequence of the Trypanothione Reductase (TR) enzyme from *Trypanosoma cruzi* was retrieved from NCBI with Uniprot accession no. P28593. After careful structural analysis and considering the reliability of the model, 1AOG (*Trypanosoma cruzi* trypanothione reductase; Tc-TR) with a resolution of 2.3 A° was selected for further investigations. The homo-dimeric protein structure with two subunits (A, B) was well characterized by different scoring methods and 0.3% residues in the disallowed region, as depicted by Ramachandran plot, confirmed the authenticity of the model.

The identification of potential binding site(s) for the test compounds, i.e., ligands, is a vital step because it provides an insight into the active site region on the protein and gives an idea of the interacting residues, as well as the crucial interactions involved between the protein and ligands [24]. The active site for the binding of ligands was predicted using DeepFold (a deep learning-based server which works on spatial restraint-guided structure prediction [25].

Gibbilimbol B and the two synthetic derivatives, LINS03018 (**1**) and LINS03024 (**2**), were used to perform the molecular docking studies. All three test compounds were energy minimized using conjugate gradient and steepest descent methods and were docked with the target protein, i.e., Tc-TR, using AutoDock Vina [26], an open-source program for molecular docking. The docking scores were obtained in terms of a combined scoring function, which calculates the affinity or fitness of protein–ligand binding by summing up the contributions of several individual terms.

The interfaces of the protein–ligand docked complexes were analyzed to study their overall and residue level interactions. All the residue interactions between the protein and the ligand compounds obtained from molecular docking studies were plotted using Discovery studio visualizer [27].

### 2.13. Statistical Analysis

The analyses of statistical significance were performed using the software Graph Pad Prism 5 by the one-way ANOVA method and the Tukey’s Multiple Comparison test, using the *p* values between samples. The assays were repeated and the samples were reproduced in duplicate, and the representative experiments were shown.

## 3. Results

### 3.1. Hemolytic Activity

The ability of compounds **1** and **2** to induce hemolysis was analyzed by a colorimetric assay using erythrocytes of BALB/c mice. According to the obtained results, compounds **1** and **2** caused no significant hemolytic activity at concentrations between 6.25 and 200 µM (data not shown).

### 3.2. Mechanism of Action

The parameters applied for the lethal action studies described below, were determined in trypomastigotes treated at short-time incubations with compounds **1** and **2**, using resazurin to assess cell viability. Compound **1** resulted in an IC_50_ value of 230.2 μM and compound **2** in 32.06 μM.

### 3.3. Plasma Membrane Electric Potential (ΔΨ_p_)

Alterations in the ΔΨp of trypomastigotes were verified by flow cytometry using bisoxonol dye. According to the data presented in Figure 2, the fluorescence levels of the treated parasites were similar to those of the untreated group, confirming no alterations to the plasma membrane electric potential. Gramicidin (0.5 µg/mL) was used as a positive control, leading to maximum depolarization (normalized data).

### 3.4. Mitochondrial Membrane Electric Potential (ΔΨ_m_)

Variations in the ΔΨ_m_ were evaluated by flow cytometry in trypomastigotes using JC-1 dye. This fluorophore is presented as monomers with green fluorescence (BL-1) at low concentrations and forms J-aggregates with red fluorescence (BL-2) at high concentrations. Therefore, shifts in the ratio between J-aggregates and monomers (BL-2/BL-1) indicate depolarization or hyperpolarization of the mitochondrial membrane. As depicted in Figure 3, compound **1** (230.2 µM) induced a significant mitochondrial membrane depolarization (*p* < 0.006) when compared with the untreated parasites. Likewise, treatment with compound **2** (32.06 µM) also resulted in statistically significant depolarization (*p* < 0.05). The oxidative phosphorylation uncoupler CCCP (25 µM) was used as a positive control for maximum depolarization.

### 3.5. ATP Levels

When compared with the untreated parasites, compound **1** significantly decreased (*p* < 0.0023) the ATP levels of *T. cruzi* trypomastigotes (Figure 4). These results were similar to those levels obtained with the positive control, CCCP. In contrast, compound **2** induced no alteration in ATP levels at this time.

### 3.6. ROS Levels

The ROS levels on *T. cruzi* trypomastigotes were monitored using the H_2_DCFDA dye using a spectrofluorimeter. According to the results depicted in Figure 5, compound **1** resulted in a significant reduction in ROS levels (*p* < 0.0051) when compared with the untreated parasites. Conversely, treatment with compound **2** resulted in no alterations in ROS levels. Sodium azide increased the ROS levels and was used as a positive control.

### 3.7. Acidocalcisomes

The analysis of the acidocalcisomes was carried out in trypomastigotes using the acridine orange dye. In the cell, this fluorophore accumulates in acidic compartments such as acidocalcisomes. When the environment is alkalinized, the dye is released, leading to an increase in fluorescence. As can be noted in Figure 6, both compounds **1** and **2** induced significant alkalinization after 20 min of incubation when compared with the untreated parasites (*p* < 0.003). A similar profile was exhibited by the positive control nigericin, a K^+^/H^+^ exchanger that alkalinizes the acidocalcisomes.

### 3.8. Intracellular Calcium Levels

Disturbances in Ca^2+^ levels were determined in trypomastigotes using the florescence intensity of Fluo-2 AM dye. It is possible to observe in Figure 7 that neither compound **1** nor **2** showed significant alterations in calcium levels when compared to the untreated control. Treatment with nonionic surfactant 0.5% Triton X-100 (positive control) resulted in the maximum Ca^2+^ release.

### 3.9. Nitric Oxide Response of Macrophages Infected with T. cruzi

To evaluate the possible activation of host cells induced by compounds **1** and **2**, the nitric oxide (NO) levels of trypomastigotes-infected macrophages were analyzed using the Griess reaction in a spectrofluorimeter. Our data demonstrated that after 48 h of incubation with the compounds (3.75 to 30 µM), no NO level alterations were detected when compared with the untreated cells (data not shown). LPS was used as the positive control to obtain maximum NO levels.

### 3.10. Protein Modelling, Binding Site Identification, Molecular Docking Studies, and Analyses of Crucial Interactions

To investigate how the tested compounds dock with potential vital enzymes involved in energy metabolism in *T. cruzi*, the model of the protein is the primary requirement. For modelling purposes, the protein sequence of the trypanothione reductase (TR) enzyme was chosen as one potential target protein of *T. cruzi*. TR is an essential key enzyme of the unique trypanothione-based thiol metabolism of the trypanosomatidae and a promising target for the development of selective inhibitors. For the following work, we used completely automated servers based on artificial intelligence and deep learning algorithms to generate and analyze the computational data. 

The three-dimensional structural model of TR from was used from PDB (ID: 1AOG) to elucidate the fine structural details and to proceed further with the molecular docking studies. The rigid loop conformations in the model were responsible for stabilizing the position of helix, which is further responsible for the wider active site for optimum binding of the ligand molecules. As shown in Figure 8, the homo-dimeric protein consists of two subunits and accommodates α-helices, β-strands, and β-loops, which determine the overall secondary structure of the protein. 

Identification of the binding site is essential for structure-based virtual screening of compound libraries. To access the binding cavity here in *T. cruzi* TR, the TR moiety mostly contacted amino acid residues in the deep cleft present at two different positions on each subunit. As shown in Figure 9 the residues from both the subunits—GLY14 SER15 GLY16 GLY17 GLY51 THR52 CYS53 VAL56 GLY57 CYS58 LYS61 ALA160 SER161 GLY162 SER163 SER179 PHE183 PHE199 ILE200 GLU203 PHE204 ARG288 ILE325 GLY326 ASP327 MET333 LEU334 THR335 ALA338—formed the main region of both binding sites and were conserved. The conserved nature of the active site residues gives us confidence in the relative accuracy of the model. Two binding pockets, i.e., the active sites, were identified and the same were validated by the targeted docking of the three compounds into the protein.

Molecular docking is a significant step in the drug design process and is used to assess the binding ability of the test compounds to the target protein. The three test compounds were docked with Tc-TR using AutoDock Vina to evaluate their binding affinity and to investigate how precisely the ligands were docked into the binding region at the protein surface. All three compounds successfully docked and exhibited favorable binding interactions with Tc-TR. The structural and biological properties of the compounds are detailed in Table 2. All three docked compounds are shown in Figure 10 with their interactions and best docked pose were selected based on the docking scores. The three compounds showed good interactions with some of the substrate-binding amino acids, such as GLY14 SER15 GLY16 GLY17 GLY51 THR52 CYS53 VAL56 GLY57 CYS58 LYS61 ALA160 SER161 GLY162 SER163 SER179 PHE183 PHE199 ILE200 GLU203 PHE204 ARG288 ILE325 GLY326 ASP327 MET333 LEU334 THR335 ALA338, and fitted well in the active site cavity of the protein. The crucial interactions of all three compounds plotted by Discovery Studio visualizer are shown in Figure 11.

The interaction analyses performed for the crucial interactions formed between the compounds and the protein molecule revealed the contribution of all non-bonded interactions towards the overall binding of the protein–ligand docked complex (Figure 11).

### 3.11. Mechanism of Action Proposal

From the analysis of all of the obtained results, it was possible to propose a mechanism of action for the treatment of *T. cruzi* trypomastigotes with compounds **1** and **2** (Figure 12). In summary, when crossing the plasma membrane, **1** and **2** did not cause changes in the electrical potential of this barrier (a). Once inside the parasite, the compounds induced the depolarization of the mitochondrial membrane (b), with **1** affecting the respiratory chain, as evidenced by the decrease in ATP and ROS levels (c). It is also possible to verify that both compounds induced the alkalinization of acidocalcisomes (d); however, there was no increase in intracellular calcium.

## 4. Discussion

Natural products have historically made the most important contribution to drugs against infectious diseases and cancer with the use of their prototypes for the synthesis of analogues. In a previous work, derivatives of gibbilimbol B, an alkylphenol isolated from *Piper malacophyllum*, presented an antiparasitic activity against *Trypanosoma cruzi*, targeting the parasite cell membrane [8]. An improvement in the chemical structure, explored by our group, resulted in a new series with a higher potency, but lacking the targeting ability towards membranes [9,10]. Among these synthetic derivatives tested so far, LINS03018 (**1**) and LINS03024 (**2**) demonstrated good potency and selectivity in *Trypanosoma cruzi*. Additionally, both compounds were active against the two main forms of the parasite. Firstly, the amastigotes, an intracellular form of the parasite considered the most clinically relevant form, observed mainly in the chronic phase, and second the trypomastigotes, the infective form, presented mostly in the acute phase of the disease, being responsible for transmission and disease reactivation [24].

In vitro cytotoxicity assays are commonly used to determine the safety profile of a potential drug. In particular, erythrocytes can be employed as a model to assess toxicological effects, proving compound selectivity and its action on the cell plasma membrane [28]. In our studies, neither compound **1** nor **2** promoted hemolysis of mice erythrocytes at any of the tested concentrations. This promising data reduce the possibility of toxic effects to mammalian plasma membranes, as previously observed for compound **1** in *T. cruzi* [9].

In this work, using cellular-based approaches, we investigated the potential mechanisms responsible for triggering the lethal effects in *T. cruzi*. These studies were carried out using spectrofluorimetric, luminometry, and flow cytometric analyses. This information is essential for the drug discovery studies towards a lead optimization [28]. The plasma membrane is essential for the maintenance and regulation of metabolites, ions, and nutrients, as it is responsible for the influx and efflux control of homeostasis. In mammalians, cholesterol plays a major role in the plasma membrane fluidity, while in trypanosomatids this sterol is replaced by ergosterol [29]. Therefore, the investigation of the plasma membrane as a potential site of action is an indispensable strategy to establish a compound mechanism of action. Previous studies have revealed that compounds **1** and **2** affected *T. cruzi* trypomastigotes by triggering a lethal action other than plasma membrane disruption [9,10]. However, the alterations in the plasma membrane electric potential have not been checked to date. This potential is a result from the ionic flow that produces an adequate electrochemical gradient, essential to different cellular processes such as growth and survival. Therefore, disarranges in this ionic balance can lead to failures in the transport of vital substances and to the formation of pores across the membrane [30]. In the present study, no alterations in this potential were observed after treatment with **1** and **2**.

Especially in trypanosomatids, the mitochondria are considered a promising biochemical target, which plays an important role in cellular bioenergetic metabolism. In these parasites, the mitochondrion is a single organelle and exhibits morphological differences from mammalian counterparts, especially in terms of volume and number [31,32]. Our data revealed that compounds **1** and **2** promoted depolarization of the mitochondrial membrane in *T. cruzi* trypomastigotes, with compound **1** leading to a more significant alteration.

The mitochondrial membrane potential is generated through electron transfer processes during respiratory chain function, with an efflux of protons to the intermembrane space. A proper potential is fundamental to maintain the oxidative phosphorylation and the production of ATP. Alterations in the permeability to protons can lead to a bioenergetic imbalance [31,33]. Herein, the ATP generation was verified in *T. cruzi* trypomastigotes and it was possible to note that treatment with compound **1** caused impairment of the bioenergetics system, reducing these levels. Conversely, compound **2** caused no significative alterations, probably due to the lack of effect in the mitochondrial membrane potential.

Mitochondrial respiration is also the main source of ROS in the parasite. Superoxide anions, hydroperoxides, and other species are produced by O_2_ reduction and play essential roles in cell signaling, contributing to parasite proliferation. Therefore, mitochondrial imbalance directly affects ROS production, leading to oxidative damage and interfering with different biosynthetic pathways [34,35]. The results show that compound **1** promoted a decrease in ROS production, an action that might be ascribed to the respiratory chain breakdown. On the other hand, these levels were not affected by **2**, corroborating other studies that showed no significant alterations in the mitochondria homeostasis of *T cruzi*.

Acidocalcisomes are specialized organelles essential for cellular viability and are conserved in different species, from bacteria to humans. These acidic compartments are involved in the regulation of the intracellular pH gradient and in the storage of pyrophosphate, orthophosphate, polyphosphate, calcium, and other ions [36]. Therefore, alterations in the acidocalcisomes can trigger a bioenergetics imbalance related to the disparity of phosphorus compounds, considered an alternative source of energy in trypanosomatids. These organelles can also affect calcium concentrations in the cytosol, as it possesses a Ca^2+^/H^+^ transporter, responsible for the calcium uptake [37]. Our results indicate that early treatment of parasites with compounds **1** and **2** led to significant alkalization of these organelles. This suggests that the effects of both compounds may involve the acidocalcisomes integrity.

From cell mobility and invasion to cell cycle control, the intracellular calcium level is responsible for the modulation of critical signaling pathways that control several cellular functions. In trypanosomatids, this ion is compartmentalized in three main different organelles: endoplasmic reticulum, mitochondria, and acidocalcisomes [38]. Accordingly, impairment in the mechanisms of storage and the regulation of Ca^2+^ induce high cytosolic levels and, consequently, mitochondrial calcium uptake, to prevent cell damage. However, an excessive increase in the intramitochondrial calcium level leads to the formation of high conductance channels across the membrane, resulting in depolarization of the electric potential. Thus, proper maintenance of calcium levels is essential to prevent failures in the metabolic process and cell death. Interference in calcium homeostasis has also been highlighted as an interesting approach for novel antitrypanosomal candidates [38]. In our study, considering the disturbances observed in the mitochondria and acidocalcisomes, the intracellular calcium level was investigated in *T. cruzi* trypomastigotes. No significant variations were observed in those parasites treated with compound **1** and **2**. These data suggest that the impairment of the mitochondria may not be ascribed to a calcium imbalance, but probably a direct effect in the mitochondria of the parasite.

Chemotherapeutics can also induce immunomodulatory activation of host cells. This effect can upregulate chemical mediators such as nitric oxide (NO), ROS, and cytokines, contributing to eliminating intracellular pathogens [39]. Host cell activation increases NO production, which promotes the control of parasites throughout the whole course of infection. NO acts as a second messenger, playing a key role in restraining the proliferation and dissemination of parasites [40]. We investigated the NO production of *T. cruzi*-infected macrophages after treatment with compounds **1** and **2**. The results demonstrated no upregulation of NO levels, suggesting no NO-mediated mechanism in the lethal effect of the studied compounds. However, an immunomodulatory action of **1** and **2** cannot be discharged, as chemokines and cytokines can also contribute to eliminating parasites.

Considering that the compounds were designed from structural exploitation and evaluated in phenotypic assays, the parasitic targets involved in lethal effects are yet to be defined. However, we can hypothesize that the compounds may potentially target key metabolic enzymes involved in oxidative stress. As an example, the *T. cruzi* trypanothione reductase is a vital enzyme that catalyzes the NADPH-dependent reduction of trypanothione disulfide [TS2] to the dithiol trypanothione [bis(glutathionyl)spermidine, T(SH)2]. Therefore, it is a potential target for developing new antimicrobial drug molecules [41]. In the current study, we demonstrated in silico the anti-parasitic potential of three test compounds against Tc-TR. This study also supports the promising candidature of the compounds, and the stability of the protein–ligand docked complexes was also confirmed with the help of docking scores and interaction analysis studies. The modelled protein of Tc-TR was used to gain insight into the fine structural biology details, to identify the ligand binding active sites in the modelled protein and to investigate the docking affinity of the three compounds. We used an in-house conventional drug discovery pipeline to execute the molecular docking exercise, which identified that LINS03018 (**1**) and LINS03024 (**2**) exhibited good docking energy scores and showed maximum stability and binding energy during interaction studies. These in silico parameters were correlated with the biological activity of both compounds and we were also able to show good fitness, docking scores, fit values, and interaction patterns among residues using a series of molecular docking steps. Overall, these analyses have provided useful information that is required for a proper understanding of the important structural and binding features for designing novel Tc-TR inhibitors. Further biological studies are required to test the hypothesis that these compounds target Tc-TR and/or other *T. cruzi* proteins.

Taken together, the data herein obtained reinforce our previous hypothesis that different compounds from the LINS03 series may act through different mechanisms of action, even though inspired by the same prototype. Our previous results suggested that amine derivatives from this series acted as plasma membrane disruptors [11,42], but some other compounds showed a different action. Compounds **1** and **2** were among these non-disrupting compounds [9,10], motivating further investigations. These results suggest that these two compounds have different modes of action, both of which affected the *T. cruzi* mitochondria, but caused different effects in this organelle.

## 5. Conclusions

This work described, for the first time, the lethal action of compounds **1** and **2**, from LINS03 series, in *Trypanosoma cruzi*. The obtained results are important in order to understand the interferences in parasite homeostasis and will certainly support future molecular studies of these molecules.

## Figures and Tables

**Figure 1 life-13-00663-f001:**
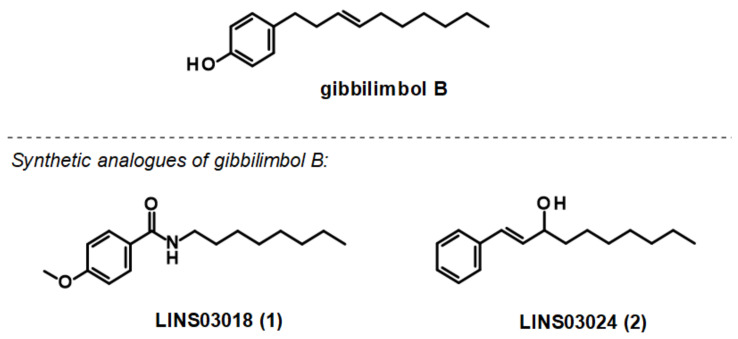
Structure of the natural prototype for gibbilimbol B and synthetic analogues previously described by the group.

**Figure 2 life-13-00663-f002:**
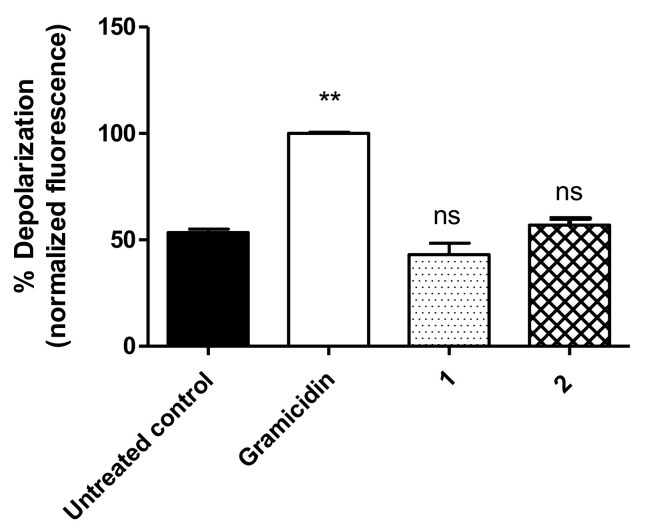
Plasma membrane electric potential (Δψ_p_) measured in *T. cruzi* trypomastigotes using bisoxonol dye after treatment with compound **1** (230.2 µM) for 3 h and compound **2** (32.06 µM) for 2 h. Gramicidin (0.5 µg/mL) was used as the positive control and untreated parasites as the negative control. Fluorescence is represented as percentage related to sertraline (100%). ** *p* < 0.0022 compared to the control; ns: not significant.

**Figure 3 life-13-00663-f003:**
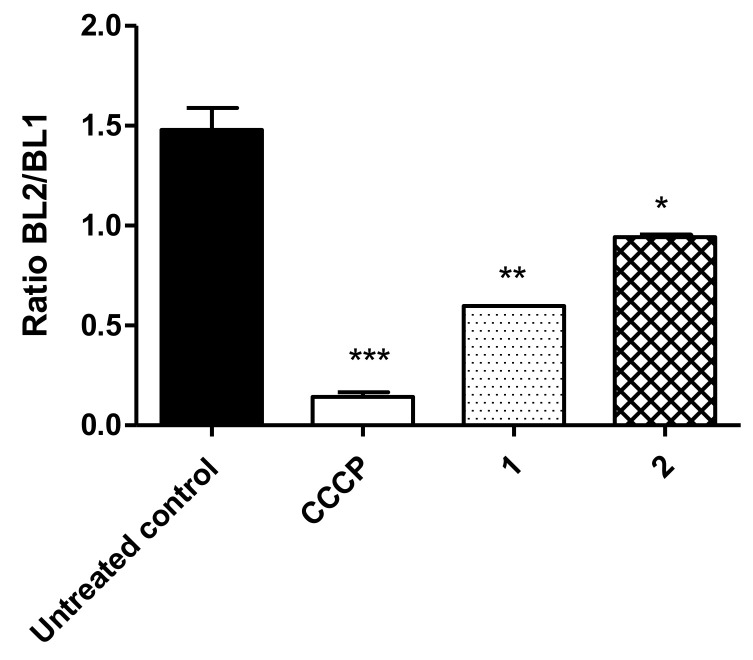
Mitochondrial membrane potential (ΔΨ_m_) in *T. cruzi* trypomastigotes measured with a flow cytometer using JC-1 dye after treatment with compound **1** (230.2 µM) and compound **2** (32.06 µM). CCCP was used as the positive control and untreated parasites as the negative control. Fluorescence is represented as the ratio between BL-2 (574 nm) and BL-1 (530 nm). *** *p* < 0.0006, ** *p* < 0.006 and * *p* < 0.05 compared to the control.

**Figure 4 life-13-00663-f004:**
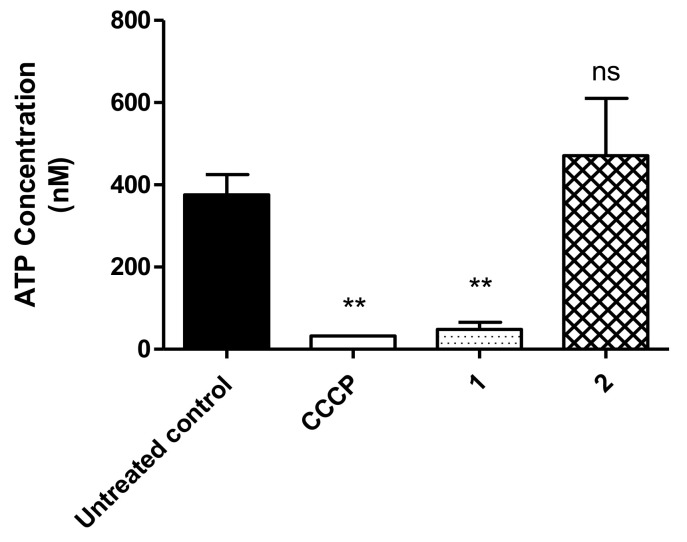
Adenosine triphosphate in *T. cruzi* trypomastigotes measured with a luminometer using the ATP Determination Kit, after treatment with compound **1** (230.2 µM) and compound **2** (32.06 µM). CCCP was applied as a positive control and untreated parasites as a negative control. ** *p* < 0.0023 compared to the control; ns: not significant.

**Figure 5 life-13-00663-f005:**
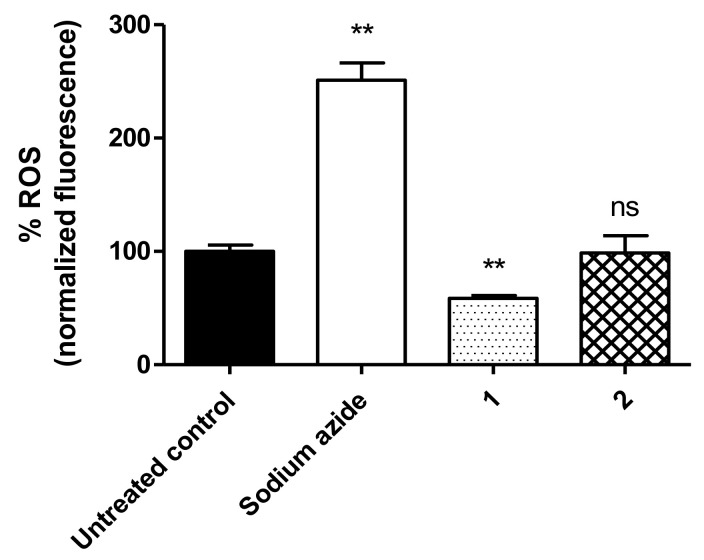
Reactive oxygen species (ROS) in *T. cruzi* trypomastigotes measured in a spectrofluorimeter using H2DCFDA (excitation of 485 nm and emission of 535 nm). The parasites were treated with compound **1** (230.2 µM) and compound **2** (32.06 µM) for 3 h. Sodium azide (10 mM) was used as the positive control and untreated parasites as the negative control. Fluorescence is represented as percentage related to the untreated control (100%). ** *p* < 0.0051 compared to the control; ns: not significant.

**Figure 6 life-13-00663-f006:**
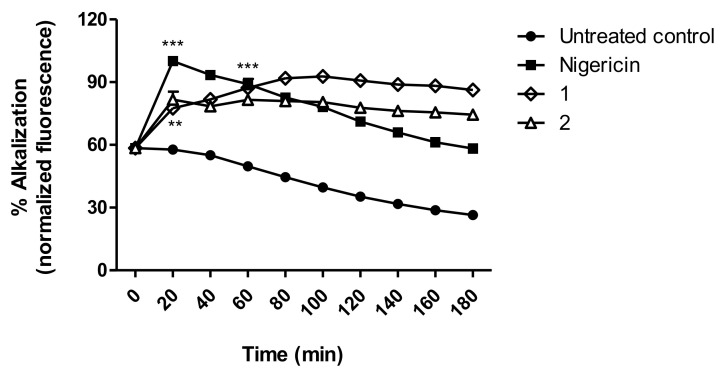
Acidocalcisomes alkalization in *T. cruzi* trypomastigotes treated with compounds **1** (230.2 µM) and **2** (32.06 µM) for 3 h. The data were obtained in a spectrofluorimeter using the acridine orange dye (excitation of 485 nm and emission of 535 nm). Nigericin was applied as the positive control and untreated parasites as the negative control. Fluorescence is represented as percentage related to nigericin in 20 min (100%). *** *p* < 0.0003 and ** 0.003 compared to the control.

**Figure 7 life-13-00663-f007:**
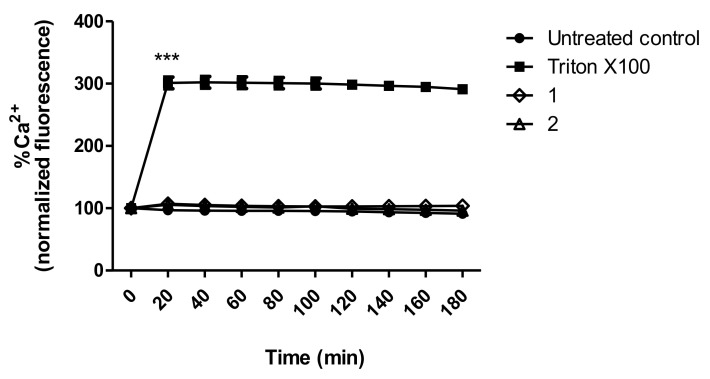
Intracellular calcium (Ca^2+^) levels in *T. cruzi* trypomastigotes treated with compounds **1** (230.2 µM) and **2** (32.06 µM) for 3 h. The data were obtained in a spectrofluorimeter using the Fluo-4 AM dye (485 nm of excitation and 535 nm of emission). Triton X-100 0.5% was used as the positive control and untreated parasites as the negative control. Fluorescence is represented as percentage related to the untreated control in 0 min (100%). *** *p* < 0.0001 compared to the control.

**Figure 8 life-13-00663-f008:**
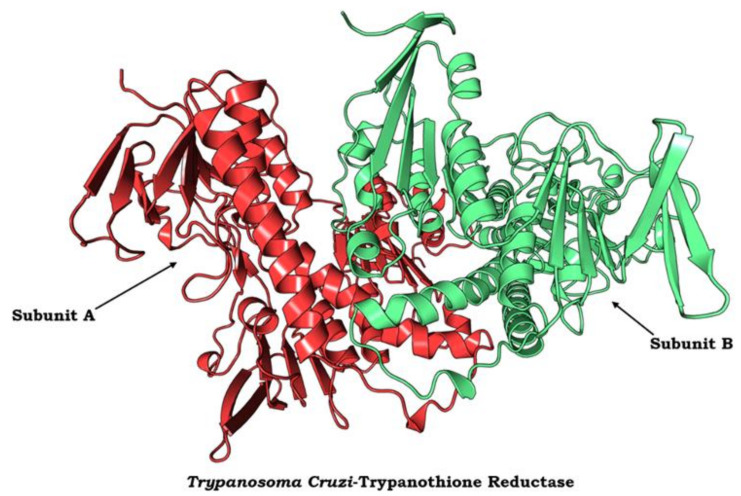
Protein model of trypanothione reductase (TR) dimer from *T. cruzi*. The homo-dimeric protein consists of two identical subunits (Subunit A in red color and subunit B in green color).

**Figure 9 life-13-00663-f009:**
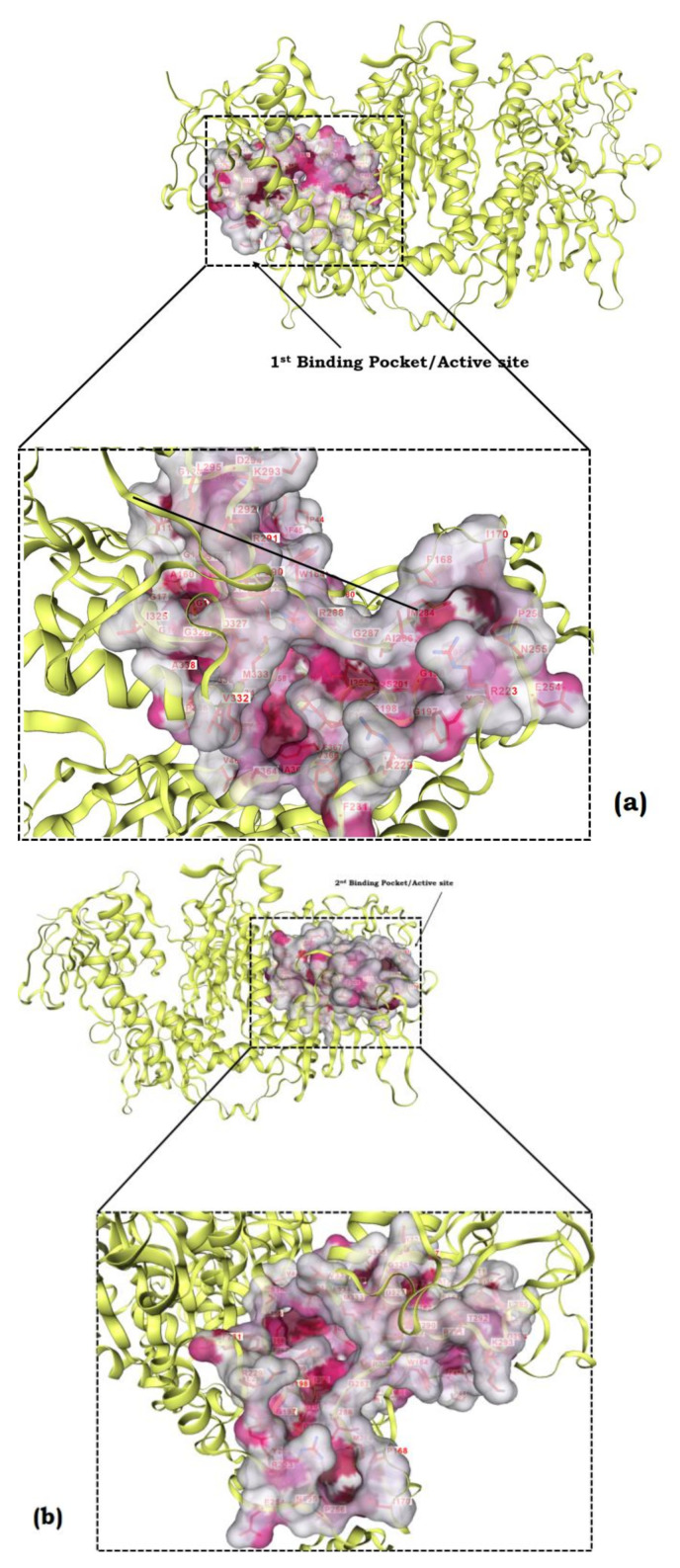
(**a**,**b**) Two binding pockets (active sites) predicted by DeepFold for T. cruzi TR. The active site region showing the conserved residues involved in protein–ligand binding and conserving the architecture of binding cavity are shown.

**Figure 10 life-13-00663-f010:**
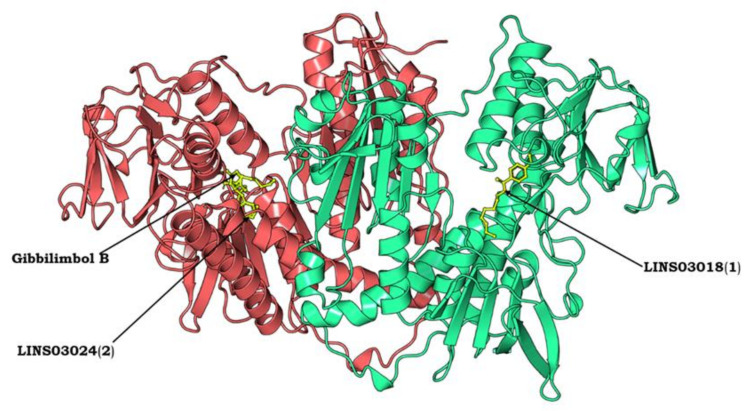
Molecular docking diagram showing all three test compounds docked into the binding cavity of Tc-TR. Gibbilimbol B and LINS03024 (**2**) are docked into the first binding pocket whereas LINS03018 (**1**) is docked into the second binding pocket. All the three docked compounds exhibit good docking scores and retain all of the conserved residues in the protein–ligand binding interaction.

**Figure 11 life-13-00663-f011:**
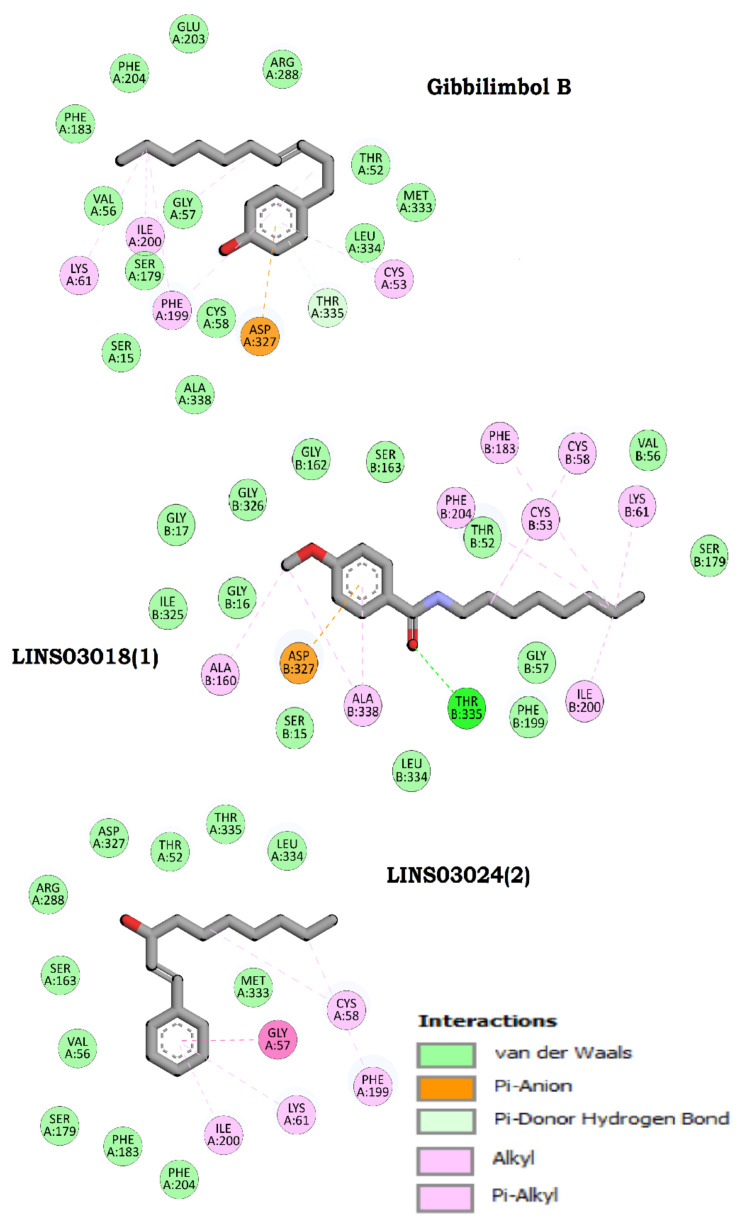
The 2D interaction diagrams of the three test compounds docked with Tc-TR, plotted using the Discovery Studio visualizer. The interactions of all three molecules obtained by molecular docking were analyzed, and all the polar and non-polar interactions are color-coded and indicated on the bottom right.

**Figure 12 life-13-00663-f012:**
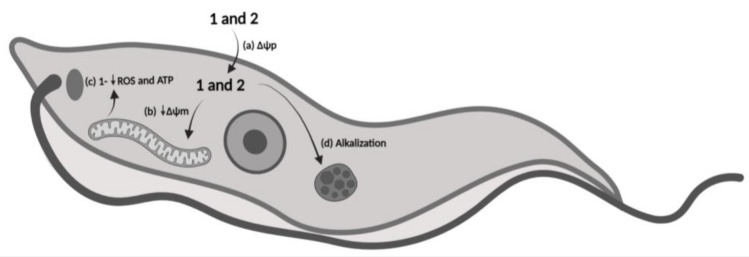
Proposed mechanism of action in trypomastigotes of *T. cruzi* after treatment with compounds **1** and **2**. (**a**) No alterations in the plasma membrane electric potential (ΔΨ_p_); (**b**) depolarization of the mitochondrial membrane electric potential (ΔΨ_m_); (**c**) only compound **1** decreases ATP and reactive oxygen species (ROS) levels; (**d**) Alkalization of acidocalcisomes with no increase in cytosolic Ca^2+^. Created with BioRender.com.

**Table 1 life-13-00663-t001:** Anti-*T. cruzi* activity and mammalian cytotoxicity of the natural prototype gibbilimbol B and its synthetic analogues.

Compounds	IC_50_ Trypomastigote (µM) ± SD	IC_50_ Amastigote (µM) ± SD	CC_50_ NCTC Cells (µM) ± SD	SI
gibbilimbol B ^1^	75.3 ± 9.2	>100	>200	ND
LINS03018 (1) ^2^	11.4 ± 5.8	5.2 ± 0.6	91.5 ± 34.8	17.6
LINS03024 (2) ^3^	9.8 ± 1.5	5.8 ± 0.1	>200	>34.5

IC_50_: 50% inhibitory concentration; CC_50_: 50% cytotoxic concentration; SI: selectivity index, given by the ratio between CC_50_ (NCTC cells) and IC_50_ in intracellular amastigotes; SD: standard deviation; ND: not determined. ^1^ Available from [11]. ^2^ Available from [9]. ^3^ Available from [10].

**Table 2 life-13-00663-t002:** The three novel compounds with their chemical structure, docking score, free energy score, and binding affinity scores.

Compound	Structure	Docking Score (kcal/mol)	Estimated ΔG (kcal/mol)	Binding Affinity (kcal/mol)
Gibbilimbol B	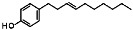	−4290.01	−7.79	5.5
LINS03018 (1)	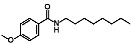	−4288.93	−7.90	6.0
LINS03024 (2)	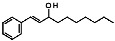	−4287.92	−7.93	5.9

## Data Availability

The data presented in this study are available on request from the corresponding author.

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
