# Peer review of "Synthetic Analogues of Gibbilimbol B Induce Bioenergetic Damage and Calcium Imbalance in Trypanosoma cruzi"

_life, 2023, doi:10.3390/life13030663_

Round 1

Reviewer 1 Report

Authors have to fulfill the following concerns

1-Better fill the IC50, CC50...etc in a table

2- Please indicate non significance on the referred columns in Figure 2, 4, and Figure 5

3- An illustration summarizing the mechanism of action of the tested compounds would be so useful

4- Consider molecular docking of the tested compounds against vital enzymes for electron transport chain and other mitochondrial enzymes

Author Response

Reviewer #1

1-Better fill the IC50, CC50...etc in a table

Authors: We appreciated all comments of Reviewer #1 and addressed them in their totality on the edited version of the manuscript where changes were highlighted in yellow. The IC50, CC50 and others were added at a new table (Table 1).

Reviewer #1

2- Please indicate non significance on the referred columns in Figure 2, 4, and Figure 5

Authors: Thank you for the comments. We modified it according to your suggestions.

Reviewer #1

3- An illustration summarizing the mechanism of action of the tested compounds would be so useful

Authors: Thank you for the suggestion. We added the requested illustration summarizing the mechanism of action of the tested compounds.

Reviewer #1

4- Consider molecular docking of the tested compounds against vital enzymes for electron transport chain and other mitochondrial enzymes

Authors: Thank you for the suggestion. We invited two colleagues from the University of Southampton (United Kingdom), Prof Ravi Kant and Prof. Myron Christodoulides, to performed an in silico molecular docking study. The new data were included as Figures 8,9,10,11 and Table 2, with the methodology and discussion.

Reviewer 2 Report

The manuscript is well written and contains valuable information. However, is necessary to clarify many points.

1. In introduction section, please describe generally the T. cruzi life cycle for better understanding of trypomastigotes and amastigotes. Also, please describe the advantages and disadvantages of tradicional drugs vs naturals drugs.

2. In materials and methods section, please describe generally the culture medias or provide the manufacturer name.

3. In results section, I don't understand why the graphics are separated. Please, merge the results into one graphic.

Author Response

Reviewer #2

The manuscript is well written and contains valuable information. However, is necessary to clarify many points.

  1. In introduction section, please describe generally the T. cruzi life cycle for better understanding of trypomastigotes and amastigotes. Also, please describe the advantages and disadvantages of traditional drugs vs naturals drugs.

Authors: We appreciated all comments of Reviewer #2 and addressed them in their totality on the edited version of the manuscript where changes were highlighted in yellow. We added the request information accordingly.

Reviewer #2

  1. In materials and methods section, please describe generally the culture medias or provide the manufacturer name.

Authors: Thank you for the comments. We described it according to your suggestions.

Reviewer #2

  1. In results section, I don't understand why the graphics are separated. Please, merge the results into one graphic.

Authors: Thank you for the suggestion. We merged them in one graphic accordingly.

Round 2

Reviewer 1 Report

I am pleased to accept the manuscript in its present form